# Evaluation of Anopheline Diversity and Abundance across Outdoor Collection Schemes Utilizing CDC Light Traps in Nchelenge District, Zambia

**DOI:** 10.3390/insects15090656

**Published:** 2024-08-30

**Authors:** Christine M. Jones, Ilinca I. Ciubotariu, Mary E. Gebhardt, James Sichivula Lupiya, David Mbewe, Mbanga Muleba, Jennifer C. Stevenson, Douglas E. Norris

**Affiliations:** 1The W. Harry Feinstone Department of Molecular Microbiology and Immunology, The Johns Hopkins Malaria Research Institute, Johns Hopkins Bloomberg School of Public Health, 615 North Wolfe Street, Baltimore, MD 21205, USA; pbacjones@gmail.com (C.M.J.); iciubot1@jhmi.edu (I.I.C.); mgebhar3@jhmi.edu (M.E.G.); 2Tropical Diseases Research Centre, Ndola P.O. Box 71769, Zambia; jamlupiya@gmail.com (J.S.L.); dmbewe495@gmail.com (D.M.); mulebam@tdrc.org.zm (M.M.); 3Swiss Tropical and Public Health Institute, Kreuzstrasse 2, 4123 Allschwil, Switzerland; stevensonj_malaria@outlook.com; 4Ifakara Health Institute, Bagomoyo P.O. Box 74, Tanzania

**Keywords:** malaria, *Anopheles funestus*, *Anopheles gambiae*, outdoor transmission, residual transmission, Zambia

## Abstract

**Simple Summary:**

In the global fight against malaria, standard mosquito control tools are designed to protect indoor spaces where people live and sleep. However, indoor residual spraying (IRS) and insecticide-treated nets (ITNs) can still leave individuals vulnerable to mosquito-borne pathogen transmission, especially outdoors, where few studies have been conducted. Even with ITNs and IRS in place, Nchelenge District in northern Zambia experiences persistently high malaria transmission. To assess the diversity and abundance of outdoor foraging female anopheline mosquitoes, light traps were used as proxy measures for mosquito host-seeking, set in three outdoor trapping schemes randomly assigned on different nights: (1) locations where people congregate outdoors at night, (2) animal pens, and (3) high-human-traffic areas, such as paths to latrines, where traps were baited with an artificial scent lure. The most important vectors of malaria parasites, *Anopheles funestus* sensu stricto (s.s.) (86%) and *An. gambiae* s.s. (2%), accounted for most of the 1087 female anophelines captured, with a highly diverse sampling of other anophelines making up the remainder. No significant difference in species diversity or female anopheline abundance was detected between trapping schemes, and malaria parasites were detected only in *An. funestus*. The challenges to malaria control in Nchelenge District may be explained, in part, by the observed outdoor foraging of anopheline mosquitoes. Light traps baited with the artificial lure demonstrated potential for outdoor collections of malaria mosquitoes in resource-poor settings.

**Abstract:**

In the global fight against malaria, standard vector control methods such as indoor residual spraying (IRS) and insecticide-treated nets (ITNs) are intended to protect inside residential structures and sleeping spaces. However, these methods can still leave individuals vulnerable to residual transmission from vectors that they may be exposed to outdoors. Nchelenge District in northern Zambia experiences persistently high malaria transmission even with ITNs and IRS in place. However, very few studies have examined outdoor vector activity. To assess the diversity and abundance of outdoor foraging female anopheline mosquitoes, CDC light traps were used as proxy measures for mosquito host-seeking, set in three outdoor trapping schemes randomly assigned on different nights: (1) locations where people congregate at night outside of the house within the peri-domestic space, (2) animal pens or shelters, and (3) high-human-traffic areas, such as paths to latrines, where traps were baited with BG-Lure^®^. A total of 1087 total female anophelines were collected over a total of 74 trap nights. *Anopheles funestus* s.s. comprised the majority of the collection (86%), with *An. gambiae* s.s. (2%) and a highly diverse sampling of other anophelines (12%) making up the remainder. *Plasmodium falciparum* parasites were only detected in *An. funestus* (1%). No significant difference in species diversity or female anopheline abundance was detected between trapping schemes. Outdoor foraging anopheline mosquitoes, including a number of infectious *An. funestus*, may partially explain the difficulty of controlling malaria transmission in Nchelenge District, where vector control is only targeted indoors. BG-Lure^®^ shows some promise as an alternative to human-baited landing catch collections in this resource-poor setting.

## 1. Introduction

In Nchelenge District of northern Zambia, holoendemic malaria transmission continues despite years of malaria interventions, including artemisinin-based combination therapies (ACTs), targeted indoor residual spraying (IRS), and long-lasting insecticide-treated bed nets (LLINs) [1,2,3]. This high year-round transmission, with rapid diagnostic test (RDT) prevalence ranging from 25% to 50%, is likely due to the marshland ecology and climate of the district [4]. The Kenani Stream runs through Nchelenge and provides breeding sites for anophelines year-round, contributing to holoendemic malaria transmission. Previous studies have shown that malaria risk is associated with proximity to streams and marshland, which are characteristic of malaria vector breeding sites [3,5].

*Anopheles funestus* sensu stricto (s.s.) Giles and *An. gambiae* s.s. Say are the major malaria vectors in this area, as determined by previous indoor vector collections [6]. *Anopheles funestus* is thought to contribute more to transmission than *An. gambiae*, as estimated by their entomological inoculation rates (EIRs of 71 and 7 infectious bites per person per year, respectively) [6], although it has been suggested that these are both underestimated, as many of these mosquitoes likely bite more than once to complete a blood meal [7]. An important driver of year-round transmission is the staggered peak abundances of the two major vector species: while *An. gambiae* abundance remains relatively low and constant throughout the year with some increase following the onset of the rains, *An. funestus* numbers are at their highest during the dry season, creating an environment where anthropophilic malaria vectors are numerous and active year-round [6,8]. Previous studies in the region have also established distinct spatial patterns for the two species. *An. gambiae* is generally found in low abundances across the study region but predominates near Lake Mweru during the wet season, while *An. funestus* is the most abundant vector overall, with relative abundances higher at inland sites away from the lakeshore along the stream networks [6,8].

The known diversity and abundance of other anopheline species in Nchelenge is limited, but is likely constrained, as most sampling has been conducted indoors. Only one outdoor pilot collection was conducted in the area prior to this study, in the form of an adapted barrier screen collection in 2015, which resulted in 274 female anophelines [9]. The proportions of *An. gambiae* and *An. funestus* in that collection were approximately equal, with 2.6% of samples remaining molecularly unidentified due to either failure or ambiguous results from standard molecular assays. Although barrier screen collections do not necessarily target host-seeking mosquitoes, collection of outdoor foraging vectors may help to elucidate why malaria incidence in Nchelenge remains high even with sustained high coverage of indoor-targeted vector control [10,11].

The human landing catch (HLC) is considered to be the gold standard method for collecting human-foraging anopheline malaria vectors and is one of the methods recommended for estimation of human biting rates by the WHO [12,13,14]. HLCs require personnel to capture mosquitoes as they attempt to land on intentionally exposed skin, but due to the potential risk of contracting mosquito-borne diseases for individuals, the ethics of HLCs have long been debated [15]. This, along with variability in attractiveness, high number of personnel required, and large operating costs, has led to a need for alternative sampling approaches. Although a number of exposure-free human-baited trap designs have been developed, each requires at least one person per collection point, making them laborious and costly given the large number of sampling units required for robust data analysis [16]. Centers for Disease Control light traps (CDC LTs) placed near people sleeping under bed nets are a common alternative to indoor HLCs and allow for a lower-cost option for multiple collection points simultaneously. They are the most commonly used method by malaria programs as part of their routine entomological surveillance activities. Utilizing this approach outdoors, however, is challenging. CDC LTs require an attractant or bait, such as a nearby host or CO_2_, but sources of CO_2_ (dry ice or pressurized gas canisters) are often unavailable or prohibitively expensive in many resource-poor settings [17]. Synthetic chemical attractants, like the BG-Lure^®^ (Biogents, Regensburg, Germany), are commercially available, easy to deploy, and have been developed and evaluated for attractiveness to several vector species, including *Anopheles* [18,19,20,21,22]. This lure is composed of a synthetic blend of chemicals meant to mimic human odors known to be attractive to arthropod vectors. Although designed for *Aedes* mosquitoes, studies have found that traps baited with BG-Lure^®^ increase catches of *An. gambiae* [18,20]. Therefore, the BG-Lure^®^ may serve as a substitute human attractant for outdoor collection schemes where other commonly used approaches are not feasible.

The Southern and Central African International Centers for Excellence in Malaria Research (ICEMR) engages in research in Nchelenge District in northern Zambia to understand the dynamics of malaria transmission and potential limitations of control measures. The primary purpose of the study presented here was to determine the diversity and abundance of anophelines foraging outdoors that may contribute to malaria transmission. The secondary objective was to compare anopheline catches of three outdoor-deployment schemes utilizing CDC LTs.

## 2. Materials and Methods

### 2.1. Study Site and Sampling Scheme

This study was conducted over an 8-day period in August 2016, during the dry season in Nchelenge District, when indoor anopheline population densities are at their highest. Eight households were selected from previous ICEMR data that indicated high abundance of anophelines caught by CDC LTs (John W Hock Ltd., Gainesville, FL, USA) set indoors. These households were divided between inland and lakeside locations (Figure 1) as previously defined [6,8]. CDC LTs with three different outdoor-deployment schemes were randomly rotated through the households for eight consecutive days. All CDC LTs were set within the peri-domestic space (defined as within 10 m from the main residential structure). The deployment schemes were as follows: (Scheme 1) CDC LTs set outdoors near where people gather at night outside of the house (such as near cooking structures or shared common areas) (24 trap nights), (Scheme 2) CDC LTs set at animal pens or shelters (24 trap nights), and (Scheme 3) CDC LTs set near high-human-traffic areas, such as paths to latrines, and baited with an artificial human-analogue bait (BG-Lure^®^) (25 trap nights). Household metadata were recorded, including number of household inhabitants, number and use of ITNs within the household, presence/absence of IRS in the household, location details of the trap, type and number of animals kept, as well as household socio-economic status, such as household education levels, house construction characteristics, and household geolocation.

### 2.2. DNA Extraction and Species Identification

Collection bags from light traps were collected each morning and transported to the field lab in Kashikishi, Zambia, where mosquitoes were killed by freezing. Mosquitoes were sorted and morphologically identified by species using standard keys [24,25], and then placed individually into microcentrifuge tubes containing silica gel desiccant, which were stored and transported to the Johns Hopkins Bloomberg School of Public Health (BSPH) at ambient temperature for molecular processing. DNA extractions were performed on the abdomens for each individual mosquito using a modified salt extraction protocol [26]. Abdominal DNA was subjected to species complex-specific PCR assays, and an internal transcribed spacer region 2 (ITS2)-based PCR when no specific PCR was available for a morphological identification or when species complex-specific results were unclear or null [27,28,29]. Specimens from a single trap (Scheme 1) were lost during transit to the BSPH and therefore did not undergo molecular analysis, although the collection count and morphological identifications were recorded. This brought the total number of female anophelines that were molecularly processed to 790 from a total of 1087 collected.

For samples yielding ambiguous results from the ITS2 PCR, as well as samples not morphologically identified as *An. funestus* or *An. gambiae*, a Barcode of Life (BOL) portion of the cytochrome oxidase I gene (COI) was amplified and sent to the Johns Hopkins Medical Institutions Synthesis and Sequencing Facility for sequencing [29,30,31,32]. Forward and reverse sequences were imported into Geneious (Biomatters, Auckland, New Zealand) version 11.1.5 (https://www.geneious.com (accessed on 15 August 2024)). Low-quality ends were trimmed, and the resulting sequences were aligned to generate a single consensus sequence for each sample. For samples where one read failed, the single high-quality trimmed read was used instead for further analyses. Corresponding sequences of COI from NCBI (N = 140) were aligned with COI sequences from outdoor samples, and phylogenetic analyses were conducted as previously described [32]. Additional sequencing of ITS2 was conducted on at least one representative from each COI-determined phylogenetic group, and sequences were compared using BLASTn against the NCBI non-redundant nucleotide database for anopheline species identification and verification [33]. Hits with a high percentage of query coverage (>70%) and a high percentage of nucleotide identity (>90%) were considered sufficient to confirm a match. Molecular analyses were combined with morphological data to generate tentative species and other taxonomic identifications. COI sequences are available from GenBank under accession numbers MK016543–MK016657. ITS2 sequences are available under accession numbers MK592014–MK592096.

### 2.3. Parasite Detection

Heads and thoraces of mosquitoes were homogenized in a grinding buffer and were split into two portions for subsequent ELISA and PCR analysis [30]. The Malaria Research and Reference Reagent Resource Center’s (MR4) circumsporozoite protein (CSP) ELISA protocol was used to detect *Plasmodium falciparum* Welch sporozoites present in the head–thorax homogenates. ELISA-positive samples were confirmed by qPCR as previously described [30,31].

### 2.4. Statistical Models and Analysis

The R packages lme4 and glmmTMB were used to fit models for this study. Interval estimates for infection rate were conducted using base R, and plots were generated using ggplot2 [34,35,36,37]. The basic structure of the models for both mosquito counts and diversity was decided based on likelihood ratio testing and comparing results from the R package DHARMa [38].

The first a priori question was: Does the expected number of mosquitoes in a given trap change as a function of the deployment scheme (animal, human, or BG-Lure^®^)? Remaining consistent with previous studies, location (lakeside or inland) was included as a fixed effect in tests as a potential confounding variable, and household was included as a random effect to account for repeated measures at each household. Overall variance was much greater than the mean for counts in this study and 31% of traps had zero captures. A negative binomial distribution fit the overall data more appropriately than an overdispersed Poisson model, and no added benefits were seen with adding a zero-inflation term to the negative binomial model.

The second a priori question was: Does the diversity of populations of anophelines caught in any given trap differ as a function of the trap deployment (animal, human, or BG-Lure^®^)? Absolute number of species per trap was used as the response variable, and location was added as a fixed effect with household as a random effect. For diversity, a Poisson model with overdispersion was found to give a better fit than negative binomial.

The trap that had specimens that were lost in transit was dropped from diversity and species distribution analyses. Thus, for the diversity models, the number of trap nights per treatment was 24 for the human-attractant context, 25 for the BG-Lure^®^ traps, and 24 for the animal-attractant traps (N = 73 traps total). Since abundance models rely simply on the count of female anophelines, the full anopheline count was still used for abundance statistics (N = 25 human-attractant and N = 74 total traps) for abundance models.

For both count and diversity, the final models were compared using cAIC values in the model.sel function in the R package MuMIn [39]. A version of R^2^ developed for generalized linear mixed models (GLMMs) was used to estimate the marginal and conditional pseudo-R^2^ of each model, as implemented in MuMIn’s r.squaredGLMM function using the trigamma function [39]. The R package DHARMa was used for fitted GLMM model diagnostics [38]. All abundance and diversity models tested are listed in the Appendix A. Confidence intervals were estimated from mixed models using the bootstrap method in the confint.merMod function from lme4 and compared to R’s confint function estimates from corresponding models with household-level random effects excluded [34,36].

Chi-squared tests were used to compare the species-specific distributions for the five most abundant species among collection schemes to an expected equal distribution of female anophelines. For species with low numbers (*An. gambiae* and *An. squamosus* Theobald), Fisher’s exact test was used to compare counts from each type of attractant to the expected abundance (one-third of the total abundance for that species).

## 3. Results

Over the course of the 8-day study, 1087 female anophelines were collected. Of the 790 that were processed at the Johns Hopkins Bloomberg School of Public Health, 747 (95%) were molecularly confirmed to species. The remaining samples did not amplify on any attempted PCRs. Of the 747 species-confirmed specimens, the majority of these were *An. funestus* s.s. (86%), with *An. gambiae* s.s. (2%) and other anopheline species (12%) making up the remainder.

### 3.1. Estimates of Infection and Transmission Rate

Six of 724 specimens tested by CSP ELISA (and confirmed by qPCR) were positive for *P. falciparum* sporozoites, all detected in *An. funestus* s.s. Based on these results, the estimated infection rate for *An. funestus* for this collection was 1.2% (95% CI: 0.24%, 2.1%). Mean catch per household per night was used as a proxy foraging rate in entomological inoculation rate calculations as previously described [6]. The mean confirmed *An. funestus* specimens per household per night (n = 7.5) was multiplied by the infection rate, which generated an estimate of 0.09 infectious bites/person/night (ib/p/night) in Nchelenge during mid-August 2016, or an estimated 16 ib/p/6 months.

### 3.2. Models of Anopheline Abundance

Models from hypothesis testing with the highest likelihood estimates (<5 delta cAIC) indicated that location (lakeside or inland) was the most appropriate predictor of anopheline count in any given trap and the effect sizes for location in each model are highly concordant. The model with best fit by cAIC was:log (expected count) ~ (location) + (deployment) + (1|HH)

This location + deployment model (trap scheme) accounting for household-level variation predicted that (1) traps inland were expected to catch 15× (95% CI: 6.2×, 35×) more female anophelines than traps lakeside (*p* < 0.0001), and (2) neither the human-attractant deployed traps (95% CI: 0.615, 3.07) nor the BG-Lure^®^ traps (95% CI: 0.241, 1.28) had statistically different abundance (*p* values = 0.41, 0.18 respectively) from the animal pen-deployed traps. Additionally, deployment was not a statistically significant predictor in any of the tested models (Appendix A).

When abundance was analyzed to examine the impact of date of collection, the second day had statistically (*p* < 0.0003) fewer female anophelines than the first study day, and the ninth day also had fewer females, though this was not statistically significant (*p* = 0.06). A date-level random effect did not provide a significant improvement from the model. Since date was not an improvement and risked overparameterizing models, it was discarded from further analysis.

### 3.3. Models of Anopheline Diversity

There was no evidence to suggest that where traps were deployed significantly impacted the diversity of anophelines caught, as it was not a statistical predictor in any of the models tested (Appendix A). However, household location was a statistically significant predictor (*p* value, 0.0001) of diversity by itself, with 2.9× (95% CI: 1.8×, 4.8×) more anopheline species seen in traps inland compared to those set at the lakeside. The location-only model of diversity had a conditional R^2^ of 0.23 and a marginal R^2^ of 0.38. The best model by cAIC was:log (expected # species) ~ (location) + (eaves) + (1|HH)

Location was statistically significant (*p* = 0.001), with 2.2× (95% CI: 1.4×, 3.8×) more species inland than lakeside. This model also showed that open eaves predicted approximately 2.5× (*p* < 0.1, 95% CI: 1.0×, 12×) more species per trap, though this was not significant. Marginal and conditional R^2^ were 0.27/0.40. Since only one of the eight tested households had closed eaves, bias in this limited sample cannot be discounted.

### 3.4. Species Distribution

The five most abundant anopheline species were assessed in terms of distribution among both deployment scheme and site, as shown in Figure 2 and Figure 3. All *An. funestus* and *An. gambiae* specimens were molecularly confirmed as sensu stricto for their respective species groups. Of the five species, only *An. gambiae* had a statistically significant (*p* = 0.002) difference in abundance between deployment schemes (Figure 2), with most (N = 11/14) caught in traps set near where people congregate. *An. funestus* was caught equally in all traps, regardless of trapping scheme.

Distinct spatial patterns for many anopheline species are displayed in Figure 4. To account for the dramatic differences in overall abundance between lakeside and inland locations, the relative contributions of each species to either the lakeside or the inland collections were compared (Figure 3). *Anopheles gambiae* was equally common in both inland and lakeside collections, as was *An. squamosus*. *Anopheles funestus* comprised a significantly higher proportion of inland collections than it did of lakeside collections. All *An. gibbinsi* Evans (N = 29) were collected inland.

*Anopheles coustani* Laveran comprised a higher proportion of lakeside collections than inland. The complex clade structure of *An. coustani* has been explored elsewhere [30]. Ciubotariu et al. (2020) identified two distinct phylogenetic groups among *An. coustani* samples. Figure 5 illustrates that there is a statistically significant difference between the proportions of groups 1 and 2 found inland versus lakeside.

## 4. Discussion

This study found no statistically significant difference in mosquito abundance or diversity between outdoor collections with CDC LTs placed in areas where humans congregate, near animal pens, or traps baited with the BG-Lure^®^/human-analogue attractant. The relatively small sample and limited collection period (N = 74 trap nights in August 2016) of this study may have underpowered detection of smaller differences between the synthetic bait and attraction to either animals or humans in the proximity of the trap. However, this remains an encouraging finding for potential use of the BG-Lure^®^ as an approximate outdoor equivalent of standard human-baited indoor CDC LT collection, especially in regions where CO_2_ is unavailable as a mosquito attractant or where HLCs are not possible due to ethical, logistic, or cost constraints.

Collections revealed species-specific patterns of behavior that can be interpreted within the context of known species bionomics and general anopheline biology. *Anopheles gambiae*, for example, was found mainly in CDC LTs placed near locations where people congregate, which reflects its known anthropophily [7]. In contrast, other patterns of behavior deduced from the collections were surprising. For example, *An. funestus* did not show the expected bias toward human attractants and was found equally across all trap deployment schemes. This supports prior reports of animal blood meals associated with *An. funestus* s.s. and may also reflect the sheer abundance of this species across the collection area or reflect the relatively small sample [8]. The overall abundance of *An. gambiae* was lower than that of other anopheline species, which is in contrast to other studies that have shown they make up ~10% of indoor CDC LTs in this area [6]. In addition to skewing the expected proportions of species caught in Nchelenge, many of the anophelines collected in these outdoor-deployed traps, such as *An. coustani*, *An. gibbinsi* and *An. squamosus*, had not been previously documented in this region of Zambia.

*Anopheles coustani* was among the most abundant species in these collections after *An. funestus* and likely exists as a species complex. Therefore, our finding of genetic structure amongst *An. coustani* specimens is not unexpected [30,40,41]. However, that the data support biological differences among the *An. coustani* clades (in the form of spatial distribution differences) is surprising, especially for such a small sample set. The statistically significant difference in lakeside vs. inland distributions of *An. coustani* clades reported here may be due to cryptic populations (within what is currently recognized as a single taxon) or due to additional biological or environmental constraints that are not currently understood. This finding is significant in the context of *An. coustani* as an emerging malaria vector, since understanding the biological limits of a species is critical to vector control [42,43].

The occurrence of *An. gibbinsi* (previously recognized as “*An. sp. 6*”) has been corroborated by more recent work, and this taxon has been implicated as a vector of malaria in other regions of eastern Africa [44]. The distribution of this species predominantly at inland locations may be indicative of ecological limitations or preferences that may be exploited for both surveillance and control. These observations further demonstrate the value of expanding surveillance activities outdoors.

### Limitations

Though it is likely that lakeside vs. inland differences in abundance and diversity are due to ecological differences and biological suitability for mosquitoes, it is possible that socio-economically driven changes in behavior and environment (such as types and number of animals kept, presence or absence of open wells, and household coverage of vector control) between localities could contribute to these differences. A larger study focused on such factors in design and implementation would be required to elucidate the impact of these elements.

Households in this study were selected based on prior collections that identified these locations as high-yielding. Though this non-random sampling strategy is unlikely to bias comparison results between the trapping schemes evaluated here, caution must be taken not to overinterpret the data and make inappropriate generalizations. Household selection and the limited number of households in this study were possible contributors to/confounders of bionomics patterns, since low-yielding households may reflect different anopheline diversity or distinct foraging patterns. Sampling may have also biased the estimated infection rates and may not accurately represent district-wide trends.

## 5. Conclusions

This study is the first in the region to target scalable collection methods for outdoor foraging anophelines, where baits such as CO_2_ are not available. Trap deployment schemes utilizing easily deployed CDC LTs did not reveal a statistically significant difference in trap yield or female anopheline diversity between traps placed near animal pens, close to where people congregate, or those fitted with the BG-Lure^®^ human-analogue attractant. These results are promising, as they suggest that the use of a commercially available and easily deployed lure may be an alternative to human-proximity trap placement and HLCs.

There was an unexpectedly high diversity of anophelines collected outdoors in this study, as also reported by Jones et al. [32]. Years of longitudinal and cross-sectional studies conducted in Nchelenge using indoor CDC LTs have identified relatively little anopheline diversity, with the primary vectors *An. gambiae* s.s. and *An. funestus* s.s. dominating those collections. Although the detected *P. falciparum* parasites in this study were not found in these other species, the sampling limitations of this study, with few mosquitoes caught, preclude any assumption of their potential contribution to local malaria transmission in Nchelenge, especially as some of these species have been implicated as vectors elsewhere [11,25,43,45].

The spatial distributions of anophelines in this study are supportive of a highly complex ecology that likely contributes to the persistently high malaria transmission rate in Nchelenge. Additionally, as there were sufficient anophelines positive for *P. falciparum* (N = 6) in these outdoor foraging mosquitoes, outdoor transmission in the region should be considered and further investigation is warranted. Outdoor transmission in Nchelenge District would undermine programmatic control efforts that only target vectors indoors.

## Figures and Tables

**Figure 1 insects-15-00656-f001:**
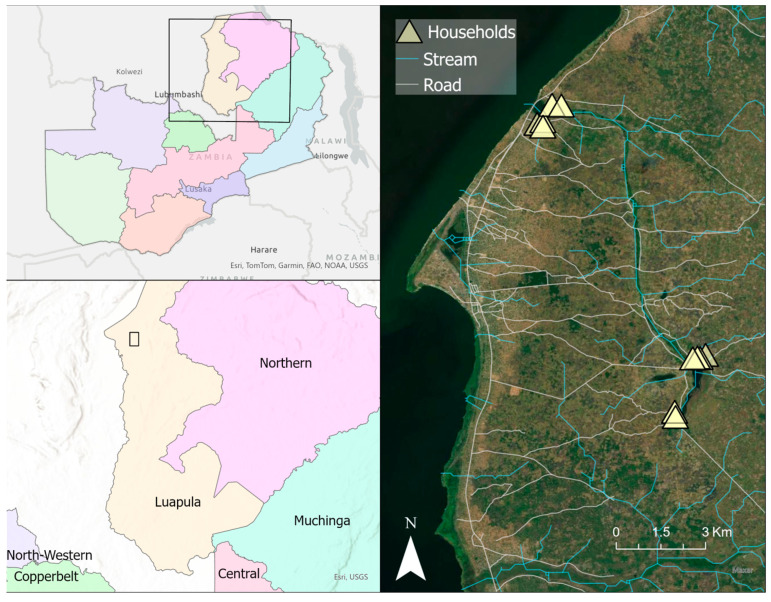
Map illustrating the location of the study site in Luapula Province in northern Zambia, set on the eastern shore of Lake Mweru. Top left panel; rectangle illustrating regional location of study area in Zambia. Lower left panel; rectangle illustrating study area in Luapula Province detailed in right panel. Triangles in right panel show the relative location of “lakeside” (**upper**) and “inland” (**lower**) household clusters. Map was generated with ArcGIS v10.6 (ESRI, ArcGIS, Redlands, CA, USA) using geocoordinates from study households [23]. Provincial border shapefiles were obtained from GRID3 Data Hub (https://data.grid3.org/datasets/d27357c640394f11943316e36cebaba3_0/about (accessed on 15 August 2024)).

**Figure 2 insects-15-00656-f002:**
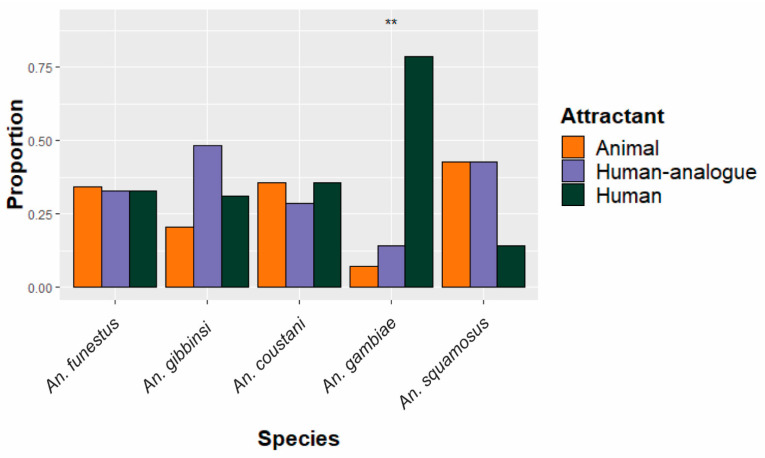
Species-specific proportion of specimens collected in traps among the three different schemes. Equal distribution among attractants would be expected if attractants had no impact on species-specific abundance. The five most abundant species are shown. ** Proportions statistically significantly deviate (*p* < 0.01) from the expected 1/3 in each attractant condition.

**Figure 3 insects-15-00656-f003:**
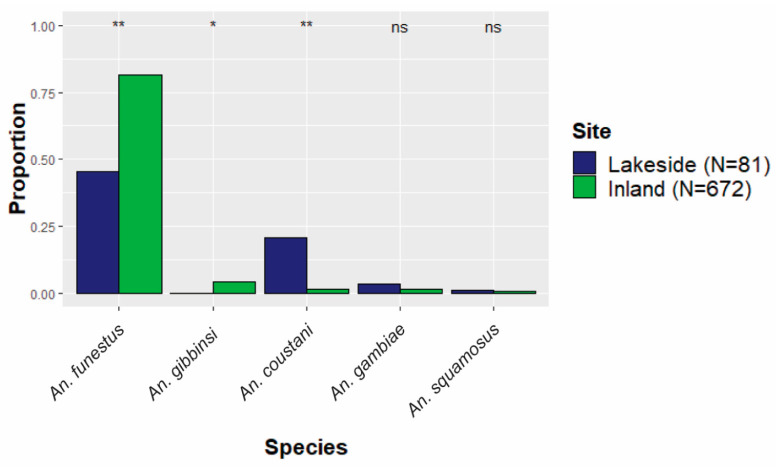
The proportion of each species relative to the total collections either inland or lakeside. The five most abundant species are shown. ** Statistically significant difference (*p* < 0.01) between lakeside and inland proportions. * Moderate difference (*p* < 0.1). ns indicates no statistically significant difference.

**Figure 4 insects-15-00656-f004:**
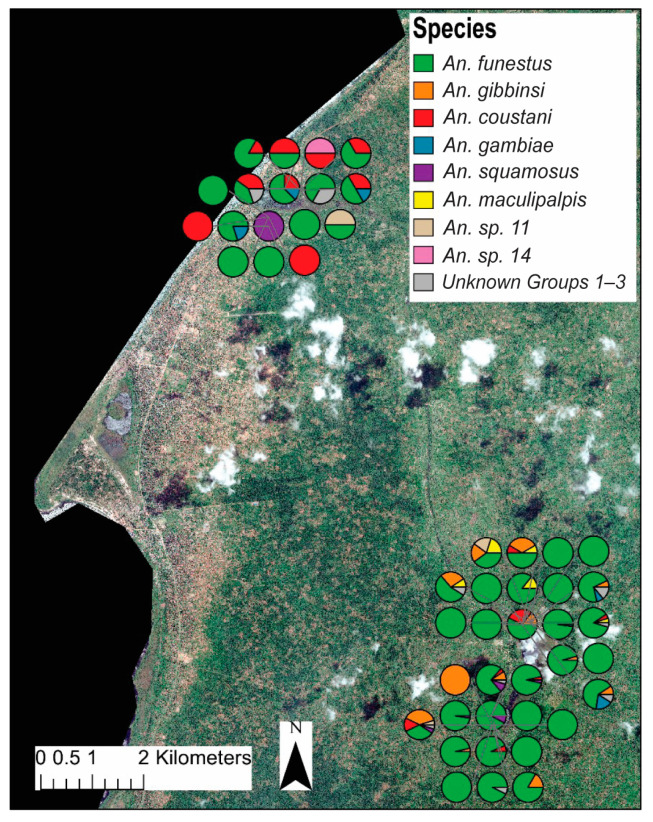
Map showing relative household/trap location and proportion of each species to total female anophelines collected per trap. The upper cluster of traps are classified as “lakeside” while the lower are considered “inland”. Map was generated with ArcGIS v10.6 (ESRI, ArcGIS, Redlands, CA, USA) using geocoordinates from study households [23].

**Figure 5 insects-15-00656-f005:**
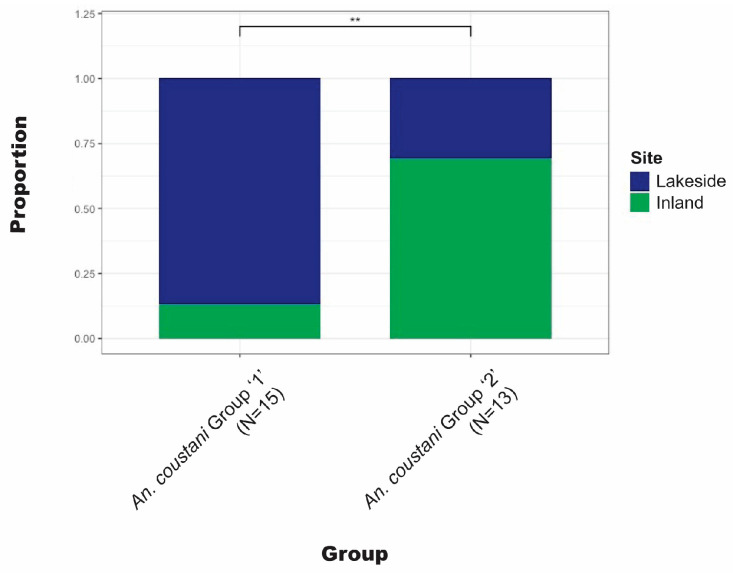
Comparison of the proportion inland vs. lakeside for each *An. coustani* clade. ** Statistically significant (*p* < 0.01) difference between clades.

## Data Availability

The COI dataset used in this study is available from GenBank (https://www.ncbi.nlm.nih.gov/genbank/ (accessed on 15 August 2024)) under accession numbers MK016543–MK016657. The ITS2 sequences are also available on GenBank under accession numbers MK592014–MK592096.

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
