# Peer review of "Evaluation of Anopheline Diversity and Abundance across Outdoor Collection Schemes Utilizing CDC Light Traps in Nchelenge District, Zambia"

_insects, 2024, doi:10.3390/insects15090656_

Round 1

Reviewer 1 Report

Comments and Suggestions for Authors

Revisions on the manuscript entitled: Evaluation of anopheline diversity and abundance across outdoor collection schemes utilizing CDC light traps in Nchelenge District, Zambia

Insects - MDPI

Simple Summary

Lines 19-20. This sentence should be rephrased for better clarity, structure is somewhat disjointed, making it difficult to follow the intended meaning. 
Suggestion: “However, these methods can still leave individuals vulnerable to mosquito-borne transmission, especially outdoors where few studies have been conducted.”

Line 25. Incomplete sentence, "of the" what? Please correct.

Line 27. Where is says: “Anopheles funestus s.s. comprised the majority of the collection”. It would be interesting to add here the percentage.

Suggestion: "Of the 1,087 total female anophelines collected, the most important vectors of Plasmodium parasites: Anopheles funestus (%) and An. biae (%) comprised the majority of the collection with a highly diverse sampling of other anophelines making up the remainer.”

Line 27. It is important to mention here that these species are known vectors of the etiological agent of malaria. 

Abstract

Line 46. I noticed that the percentages were placed here. The abstract and the simple summary are basically the same. One should exclude the other, if possible. 

Introduction

Line 114. I would suggest deleting this part, as it makes it seem like a comparaion will be done making the sentence incomplete.

Material and Methods

Line 126. How long did the collections last? One month?

Line 132. How many CDC total? 

Suggestion: " A total of XX CDC traps were set..."

Line 184. Are the mosquitoes’ vectors of only this specific species of etiological agent? Were the mosquitoes tested for other species of plasmodium or for Plasmodium genus to later on see if it belonged to falciparum or a different species?

Discussion

Line 309-310. Except for An. gambiae which had a statistically significant (p = 279 0.002) difference in abundance between deployment schemes, with most (N=11/14) caught in traps set near where people congregate. 

Line 312. What was the collection period?

Line 322. Where  it says: “Anopheles 320 gambiae, for example, was found mainly in CDC LTs placed near locations where people 321 congregate, which reflects its known anthropophily. In contrast, other patterns of behav-322 ior deduced from the collections were surprising” This is a well known fact but it is still important to add a reference here. 

Line 330-331. Where it says: “Nchelenge, many of the anophelines collected in these outdoor deployed traps had not been previously documented in this region of Zambia.” Which ones?  "...such as An.... and An...."

Line 375. Please repharse this sentence as is seems like you are talking about the present study not another one.

Suggestion: "In the study by Jones et al. (2021), an unexpectedly high diversity of anophelines was collected outdoors in Northern Zambia ."

Reviewer 2 Report

Comments and Suggestions for Authors

1.       There are errors in word spellings and some grammatical glitches in the text. It should be linguistically checked by a person whose English is good. (For example, "abundancies" should be "abundances.")

2.       Lines 2-3; The first letter of 'anopheline' in the title should be capitalized. 'anopheline' should be revised as 'Anopheline'. And same in all text.

3.       Lines 6-7; It is not appropriate to use 'on behalf of the Southern and Central Africa International Centers of Excellence for Malaria Research' after author names. If the name of the 'Southern and Central Africa International Centers of Excellence for Malaria Research' must be cited, this should be acknowledged within the institutions supported and thanked.

4.       All species and genus names should be written out in full and the describing author should be provided the first time they are mentioned; subsequently, abbreviations should be used throughout the text. For example, on lines 27, 47: Anopheles funestus, An. gambiae s.s. Plasmodium falciparum , All genus and species names should be written in italics in text.

5.       Lines 27- 46 Summary and Abstract; If s.s. is used after a species name, it should be given where it is first used (e.g. s.s. stands for sensu stricto). Afterwards, the abbreviation s.s. can be used ( For example Anopheles funestus s.s., An. gambiae s.s.)

6.       Lines 53-54: Keywords make more sense when listed in alphabetical order.

7.       Line 60; The meaning of RDT should be clearly written in the first place. Then the abbreviation should be used.

8.       Lines 142-143; Figure 1 is very inadequate. When Figure 1 is given in the materials and methods section, it should only be given to show the general geographic features of the area. It is not appropriate to give mosquito species diversity here. In the results section, mosquito and abundance graphs should be given on a larger map.

9.       Line 149; In the DNA extraction and species identification section and before morphologically identified to species, it should first be explained how the flies caught in the traps were collected, how they were transported to the laboratory, and where/conditions they were stored before extraction. Which identification keys were us efor determination of mosquito species.

10.  There is no description in the text of the meteorological conditions in the reserach area where the traps were set. Humidity and temperature have a great influence on the flying behavior of mosquitoes. It can also affect the catching power of the traps. This should be explained in the text.

11.       What animals are in the animal shelters where the traps are located? For example, cattle, goats, sheep, dogs, etc.

12. I suggest that the references be organized according to the journal's style guide (e.g., publication dates bold/non-bold, punctuation after journal names, full italics or not, abbreviations, species names in italics, etc.).

Comments on the Quality of English Language

      There are errors in word spellings and some grammatical glitches in the text. It should be linguistically checked by a person whose English is good. (For example, "abundancies" should be "abundances.")
